# When the Locus Coeruleus Speaks Up in Sleep: Recent Insights, Emerging Perspectives

**DOI:** 10.3390/ijms23095028

**Published:** 2022-04-30

**Authors:** Alejandro Osorio-Forero, Najma Cherrad, Lila Banterle, Laura M. J. Fernandez, Anita Lüthi

**Affiliations:** Department of Fundamental Neurosciences, University of Lausanne, CH-1005 Lausanne, Switzerland; alejandro.osorioforero@unil.ch (A.O.-F.); najma.cherrad@unil.ch (N.C.); lila.banterle@unil.ch (L.B.); laura.fernandez@unil.ch (L.M.J.F.)

**Keywords:** NREM sleep, REM sleep, monoamine, noradrenaline, arousability, sleep architecture, infraslow time scale, microvasculature, sleep disorder, Alzheimer’s disease

## Abstract

For decades, numerous seminal studies have built our understanding of the *locus coeruleus* (LC), the vertebrate brain’s principal noradrenergic system. Containing a numerically small but broadly efferent cell population, the LC provides brain-wide noradrenergic modulation that optimizes network function in the context of attentive and flexible interaction with the sensory environment. This review turns attention to the LC’s roles during sleep. We show that these roles go beyond down-scaled versions of the ones in wakefulness. Novel dynamic assessments of noradrenaline signaling and LC activity uncover a rich diversity of activity patterns that establish the LC as an integral portion of sleep regulation and function. The LC could be involved in beneficial functions for the sleeping brain, and even minute alterations in its functionality may prove quintessential in sleep disorders.

## 1. Introduction

Noradrenaline (NA) is a monoamine neurotransmitter that acts in the brain and body to induce and coordinate states of wakefulness, and to facilitate adaptive behaviors in response to environmental novelty. The mammalian brainstem contains a cluster of up to seven NA-synthetizing nuclei (A1–A7) that have been anatomically identified in >80 mammals [1], from rat [2], to cat [3], to human [4]. The tightly appositioned A4 and A6 nuclei stand out as the largest, often densest, and predominant forebrain-projecting nuclei that share a common embryonic origin [5] and in which activity levels correlate with the degree of wakefulness (for review, see [6,7,8,9]). In tissue sections, these nuclei appear sky-blue because of their pigmentation with neuromelanin, a by-product of catecholamine metabolism, which gave it the name *locus coeruleus* (LC, Latin for “sky-blue spot”). The LC lies in the pontine brainstem as an anteroposteriorly extended tube with a central ventral extension along the fourth ventricle (for review, see [1,8]) and it is part of the ascending arousal systems, together with other monoaminergic and cholinergic nuclei (for review, see [10,11]). The LC provides brain-wide axonal arborizations and fine meshworks of varicose fibers that arise from a comparatively small number of NA-synthetizing neurons (thousands in rodents [12,13], tens of thousands in humans [14]). The axons from LC cells span the neuraxis from the spinal cord to the cerebellum, midbrain, thalamus, and cortex and are thought to release NA through both synaptic and non-synaptic release mechanisms (for review, see [15,16]) to regulate neurons, glial cells, and fine microvessels (for review, see [17,18,19]), stimulating wakefulness and attentional orienting (for review, see [8]), sensory processing (for review, see [20]), muscle tone (for review, see [21]), and breathing (for review, see [22]), while inhibiting sleep-promoting brain areas (for review, see [10,23]). The LC also plays prominent roles in pathological forms of arousals, commonly linked to acute stress (for review, see [24]), post-traumatic stress disorder (for review, see [25]), pain and analgesia [26], motivation and relapse (for review, see [27]), hypercapnia (for review, see [28]), and hypotension (for review, see [29]), many of which are accompanied by sleep disturbances.

Novel anatomical and physiological technologies, together with advanced behavioral measures, are about to bring fundamentally renewed insights into the LC’s functions. The LC shows a genetic and/or functional heterogeneity at multiple levels from its embryonic and evolutionary origins (for review, see [1,5]), its synaptic interactions with the pericoerulear area (for review, see [30]), its input–output connectivity (for review, see [31]), to its cellular identities and neurotransmitter release (for review, see [6,30]), neuronal ensemble formation (for review, see [32]), regulation of whole-brain states [33], brain-state-dependent firing patterns (for review, see [7,30]), and behavioral roles (for review, see [34]). The LC emerges as a dynamic and plastic assembly of functionally specialized LC neuronal subgroups that act locally or globally according to recently lived experiences, ongoing demands, and future challenges (for review, see [30,35]).

Time is also ready to complement the prevalent LC-wakefulness association with the appreciation that the LC is important for sleep. The central message of this review is that LC’s role in sleep has remained underestimated. Novel real-time monitoring and interferential approaches now start to indicate that LC contributes to sleep in fundamental ways—to its cellular functions, its micro- and macroarchitectural organization and regulation, associated behaviors, and possible roles in disease. These insights are at their very beginnings, yet they indicate that the LC could become an important factor in profiles of perturbed sleep that arise from diverse conditions. In this review, we discuss these exciting developments primarily based on animal experimentation, but we include human studies whenever they help complement available evidence. For a more human sleep-oriented recent review on the LC’s role in sleep, we refer to Van Egroo et al. [36].

## 2. The Activity of the LC in Sleep: Pioneering Studies

This chapter reviews studies from the last decades that provided evidence for a maintained activity of the LC in sleep. Quantitatively, these studies revealed that the LC unit activity was clearly lower compared to wakefulness, yet remained distinctly elevated during non-rapid-eye-movement (NREM) sleep compared to REM sleep. NREM and REM sleep are the two major mammalian sleep states, also referred to as “quiescent” and “active” sleep, respectively. These two states show distinct spectral characteristics and functions for sleep (for review, see [37]). Qualitatively, the studies summarized in this chapter suggest that the noradrenergic system appeared to be involved in the alternation of NREM and REM sleep. For these results, diverse techniques in animals and humans were used that span from electrophysiology and pharmacology to microdialysis and functional imaging. A summary of the traditional view that has emerged from these studies is shown in Figure 1 (left).

### 2.1. Animal Studies

Rat [38,39], cat [40,41,42], and monkey [38,43] recordings showed that the action potential discharge rates of LC units during NREM and REM sleep were minor compared to wakefulness. Unit activity was low during NREM sleep, but remained detectable, while it ceased during REM sleep [38,39]. However, researchers also noted that not all putative LC units reduced activity during NREM and/or REM sleep [40,41]. LC activity was also low in quiet—as opposed to active—wakefulness [39,42]. More recent studies indicate that some LC units may even be as active in NREM sleep as in quiet wakefulness and occasionally fire in bursts [44,45,46]. Furthermore, although activity during sleep states was overall low, it nevertheless was not random. For example, LC unit activity has been related to the organization [39] and termination [46] of sleep spindles, an essential NREM sleep rhythm in the 10–15 Hz frequency range originating from the thalamo-cortical loop [47]. Additionally, LC unit activity during NREM sleep preceded the cortical up-state periods from another important NREM sleep-associated slow rhythm, the cortical slow oscillation (~1 Hz) [45], and was increased during a post-learning NREM-sleep period [44].

Microdialysis allows sampling of mean concentrations of neurochemicals present in the extracellular fluid surrounding neural tissue. Microdialysis for NA revealed its levels to be lower for states of sleep compared to wakefulness across rodents, cats, and seals, yet values for NREM sleep consistently were intermediate with respect to the ones for wakefulness and REM sleep in various brain areas (for review, see [48]). This suggested that even low LC unit activity leads to detectable NA release. However, no or minor increases in cortical NA levels in response to electrical or chemical stimulation of the LC were observed at low (1–3 Hz) compared to higher (>5 Hz) stimulation frequencies [49,50,51]. The fast-scan cyclic electrochemical voltammetry technique allows millisecond-resolution assessments of NA levels with nanomolar sensitivity, but it has so far been mostly applied for high-frequency stimulation of the LC [52]. Therefore, the relation between LC unit activity and real-time NA output has remained undefined.

Jouvet’s monoaminergic theory of sleep–wake control [53] prompted examinations of the spontaneous sleep–wake cycle after lesion or pharmacological manipulation of LC and NA signaling, or after constitutive removal of genes encoding proteins involved in NA turnover. These approaches made it clear that noradrenergic activity sustains wakefulness at the expense of sleep (for review, see [7,8,9,23]). At the same time, they provided the first hints that NA signaling remained relevant for sleep. For example, neurotoxic lesions of noradrenergic LC neurons or genetic elimination of the NA-synthetizing enzyme dopamine-β-hydroxylase (DBH) altered the relative times spent in NREM and REM sleep [54,55,56]. These approaches lacked the necessary specificity in time and in the site of action to conclude about the LC’s role in regulating the timing of NREM and REM sleep. Furthermore, noradrenergic receptors are expressed both centrally and peripherally, and LC projections target both sympathetic and parasympathetic autonomic pathways (for review, see [29]). Therefore, systemic drug administration may affect sleep–wake states by acting on the autonomic nervous system. Nevertheless, these studies are part of initial evidence that monoaminergic systems, including NA, could remain active in sleep.

Particularly noteworthy are the effects of pharmacological α2-adrenergic receptor activation. These receptors are G${}_{i}$-protein-coupled receptors activated by NA in the central nervous system and periphery. In the brain, they act as both presynaptic negative autoreceptors within the LC and in sleep–wake regulatory centers to suppress NA release and attenuate postsynaptic excitability [57] (for review, see [58]). These receptors are also the target of powerful sedatives used in clinics, such as dexmedetomidine (for review see [58]). The α2 agonist clonidine suppresses the activity of the LC [59,60] but also targets pre- and postsynaptic receptors in sleep-regulatory areas (for review, see [23]). α2 agonists such as clonidine or detomidine, when applied locally in cat pontine brainstem [61] or peri-coerulear areas [62], or systemically in rat [63], suppressed REM sleep while increasing the depth of NREM sleep. The use of clonidine in humans was also found to attenuate REM sleep (see Section 2.2). These effects on sleep macroarchitecture are in line with an active LC during sleep.

### 2.2. Human Studies

The functional activity within dorsal brainstem areas, including the LC, was examined through functional magnetic resonance imaging (MRI) in sleeping healthy individuals [64]. This imaging technique uses magnetic resonance signals to detect changes in brain activity based on increases in the flow of oxygenated over non-oxygenated blood. Signal increases involving the LC were particularly prominent during NREM sleep-associated slow (<1 Hz) waves. More recently, advances in high-resolution neuroimaging techniques allow for a refined investigation of the human LC, which has raised much attention regarding its role in sleep (for review, see [36,65]). Neuromelanin’s paramagnetic properties make MRI-based anatomical measures of the LC possible to determine its location and structural integrity. Positron emission tomography can provide estimates of noradrenergic terminal density. First studies have correlated structural and functional read-outs of the LC to human sleep, finding associations between these and microarchitectural alterations in sleep [66] that are relevant in the context of neurodegenerative disorders (see Section 5.1) (for review, see [36]).

Similar to animal models, pharmacological studies in humans using α2-receptor-specific agonists provide evidence for the role of noradrenergic signaling in the timing of NREM and REM sleep. The α2 agonists clonidine or guanfacine produced a reduction of REM sleep [67] and an increase of NREM sleep [68] while the α2 antagonist idazoxan increased the time spent in wake but also reduced the time in REM sleep [68]. Furthermore, clonidine decreased peripheral NA levels during sleep [69], consistent with a suppression of an active LC during sleep. Administration of the NA reuptake inhibitors reboxetine, maprotiline, or nomifensine, for which there is evidence that they elevate peripheral levels of NA, also suppressed REM sleep [69,70]. These studies indicate that noradrenergic signaling, in part through α2 receptor activation, is a pathway for sleep control. How this signaling modulates both local LC networks and their synaptic targets to both NREM and REM sleep control centers remained open for further study.

## 3. The Activity of the LC in Sleep: Novel Insights

The development of genetically encoded sensors for free NA now makes it possible to measure its real-time dynamics with high spatial and temporal resolution [71]. It enables a direct estimation of the relative NA levels released during the natural sleep–wake cycle and how they relate to traditional LC activity measures. These sensors are G-protein-coupled-receptor-activation-based (GRAB) and are constructed from mutated α2 adrenoceptors coupled to an EGFP moiety. When expressed in vivo through viral vectors, these GRAB sensors become localized on membrane surfaces and emit green fluorescence (∼520 nm) upon blue light excitation (∼510 nm) once NA released from LC fiber binds. High (GRAB${}_{NE1h}$) and medium (GRAB${}_{NE1m}$) affinity versions of these sensors have been presented, and renewed versions keep being developed, expanding the range of sensitivity and kinetics with which measures can be taken (see http://www.yulonglilab.org/faq.html, accessed on 1 February 2022). Furthermore, a mutant version of the sensor that is not responsive to NA should be used to control for potential non-specific alterations of the fluorescence signal that can limit its interpretation. For example, local alterations in neuronal environments, such as in brain temperature or blood pressure accompany transitions between NREM and REM sleep. These could alter light scattering or biosensor properties in vivo. Two studies in mouse, one published [72], one yet to be peer-reviewed [73], have now used these GRAB${}_{NE}$ sensors to describe the real-time dynamics of free NA levels in the medial prefrontal cortex [73] and in the primary sensory thalamus [72] during the natural sleep–wake cycle. These studies report unexpectedly high levels of NA during NREM sleep compared to wakefulness. Furthermore, they observe a dynamically varying signal during states of sleep. This chapter presents the most important findings derived from these two studies. A summary of the resulting revised view on NA signaling in sleep is shown in Figure 1 (right).

### 3.1. Mean NA Levels Differ across the Sleep–Wake Cycle

The signals provided by the genetic sensor showed characteristic alterations across wakefulness, NREM sleep, and REM sleep. In the prefrontal cortex, mean NA levels during wakefulness were high but variable [73], which is consistent with the large variations in LC activity in wakefulness (see Section 2.1). During NREM sleep, the mean free NA levels became lower but still overlapped with the ones of wakefulness. During REM sleep, the levels of NA were consistently low. In the sensory thalamus, similar measures of NA even revealed that mean levels were significantly higher during NREM sleep when compared specifically to quiet wakefulness (Figure 2) [72]. Again, values were low during REM sleep in this area. These findings provide the first evidence that NA levels remain more elevated in NREM sleep in forebrain areas than what was expected based on unit measures (see Section 2.1). The expected low NA levels during REM sleep appear as a common feature across the recorded areas. The considerable discrepancy between the numerically sparse LC unit activity (see Section 2.1) and high free NA levels generated during NREM sleep shows that much remains to be learned about the mode of operation of LC neuronal ensembles in different states of vigilance.

### 3.2. NA Levels and LC Activity Fluctuate during NREM Sleep

The next notable observation found in both the thalamus and prefrontal cortex is that NA levels were not steady during NREM sleep (Figure 2A–C). Instead, these fluctuated on an infraslow timescale of tens of seconds, with an average cycle length of 30–50 s [72,73]. These fluctuations in NA levels were linked to phasic bouts of LC neuronal activity over the same intervals, as evident by correlated Ca${}^{2+}$ transients in LC somata [73]. This activity pattern points to a periodic synchronization of LC population activity on an infraslow time scale during NREM sleep [74].

The role of these recently identified fluctuations is a current topic of investigation [72,75,76] (for review, see [77]). Optogenetic modulation of noradrenergic LC neuronal activity evoked variations in the appearance of sleep rhythms and heart rate, suggesting that infraslow NA fluctuations are relevant for NREM sleep’s physiological correlates. Thus, NA released by the LC lead to a periodic clustering of sleep spindles, such that they appeared at high density when NA levels were low and they were scarce when these levels were high (Figure 2A) [72,73]. Mechanistically, sleep spindle clustering relied on the α1- and β-adrenergic receptor-mediated modulation of membrane potentials in the thalamic circuits, in which sleep spindles are generated [72]. Sleep spindles are involved in the sleeping brain’s elaboration of sensory input (for review, see [78]), which implies the LC in NREM sleep-related sensory processing (see Section 4.1).

NA fluctuations also correlated with infraslow variations in heart rate during NREM sleep (Figure 2A). The LC thus acts bidirectionally to coordinate forebrain sleep spindle rhythms with heart rate variations. Indeed, optogenetic activation of LC noradrenergic neurons disrupted the heart rate variations during NREM sleep and their anticorrelation with the spindle clustering [72,76]. Mechanistically, the coupling of LC activity to the heart rate depended on parasympathetic signaling. Likewise, parasympathetic signaling also underlies coordinated infraslow fluctuations between pupil diameter and sigma power during NREM sleep [79].

### 3.3. NA Levels Decay to Low Levels during REM Sleep

The NA levels declined in both the prefrontal cortex [73] and the thalamus [72] during REM sleep, in line with the quiescence of LC units in this behavioral state (Figure 2B). As a result, NA levels reached a level that lay below that of wakefulness and NREM sleep. This result directly and strikingly supports the proposition that REM sleep periods are relatively NA-free (see Section 5.2). The quantification of the extent and time course of this decline will now allow us to refine this proposition, in particular in terms of the relation to REM sleep bout duration.

### 3.4. NA Levels Show Characteristic Dynamics at Behavioral State Transitions

The dynamics of NA levels at moments of transition from NREMS to REMS or wakefulness showed characteristic properties. At NREM-to-REM transitions, a decrease in NA levels began ∼40 s before the onset of REM sleep (Figure 2B). This time period recalls a transitional moment of sleep that has been referred to as “intermediate sleep” in rodents [80], cats [81] and humans [82,83]. Intermediate sleep shows a mixed spectral profile combining an increase in sigma power and the density of fast spindles, while hippocampal theta rhythms appear (for review, see [78]). On the time scale of intermediate sleep, there is a cessation of LC unit activity [39,73] and the appearance of cholinergic activity in REM sleep-promoting tegmental nuclei [84,85]. The coincidence of declining NA levels with unit and spectral correlates of intermediate sleep suggests that the activity levels of the LC during NREM sleep may determine the timing of NREM-to-REM sleep transitions.

Transition periods from NREM sleep to both sustained wakefulness and to microarousals were both associated with an increase in NA levels in the prefrontal cortex that appeared to start before the transition (∼10 s) [73]. On the same time scale, there was an increase in Ca${}^{2+}$ activity of noradrenergic LC neurons that was higher for transitions to consolidated wakefulness compared to microarousals. This appeared also to be the case for NA levels at NREM sleep-to-wake transitions (Figure 2C). These alterations are in line with unit activity measures around moments of wake-up (see Section 2.1 and Section 4.1).

### 3.5. Emerging Dynamics of Other Monoamines and Wake-Promoting Neurotransmitters

In vivo measures using genetically encoded sensors showed that, in addition to NA, other monoamines and wake-promoting neurotransmitters remain high during NREM sleep. In Ca^2+^-based fiber photometric measures of spontaneous activity in the dorsal raphe, fluctuations were observed in phase relation to spontaneous brief arousals [86]. Furthermore, measures with a genetically encoded sensor for free serotonin levels revealed slow fluctuations in both the orbital frontal cortex and the bed nucleus of the *stria terminalis* during NREM sleep, and declines during REM sleep [87]. The time course of these fluctuations, and their consistent appearance at two distant brain sites, are reminiscent of the findings with NA described in this chapter. Given the rapid advance in the availability of novel sensors for dopamine [88,89] but also for other neuromodulatory transmitters involved in sleep–wake control (such as acetylcholine, [90] or hypocretin [91]), more details on the spatiotemporal map of neurotransmitter dynamics during states of sleep will soon become available. Intriguingly, transient free dopamine increases in the basolateral amygdala were just discovered as triggers for NREM-to-REM sleep transitions [92].

## 4. The Role of the LC in the Regulation of Sleep and Sleep Functions

This chapter builds on the newly revealed real-time dynamics of NA levels described in Section 3. It aims to review how these findings advance insight and motivate experimentation in the quest for the functional roles of the LC during sleep.

### 4.1. LC as Part of Sensory Arousal Circuits during NREM Sleep

Pioneering recordings from LC units found that these respond with a short latency to stimuli from different sensory modalities [38,42,43,93,94]. Increases in LC unit discharge rates also preceded spontaneous, unsolicited awakenings from NREM sleep [38,39,94]. Moreover, activation of LC through electrical, opto-, or chemogenetic stimulation elicited transitions from sleep to wakefulness [46,95,96] and recruited whole-brain networks involved in salience processing [33]. Acute knockdown of DBH specifically in LC neurons disrupted sleep-to-wake transitions elicited by optogenetic LC stimulation, confirming the importance of NA signaling for wake-ups [96]. Given LC’s powerful capacity to drive sleep-to-wake transitions, LC activity might be involved in sensory-induced sleep–wake transitions.

Indeed, Hayat et al. [63] showed a causal link between the levels of ongoing LC activity during NREM sleep and the probability of sensory stimulus-evoked awakenings. Mild optogenetic LC stimulation lowered the auditory arousal threshold, whereas inhibiting LC heightened it. In line with this, the natural infraslow fluctuations of LC activity during undisturbed NREM sleep coincided with variations of auditory and somatosensory arousability [75,76]. Furthermore, spontaneous brief arousals from NREM sleep in mice were most frequent at moments of low spindle density, when LC activity is high [72,75].

The exact roles of the LC in the cognitive, motor, and autonomic aspects of arousal remain to be determined. As LC neurons are activated by sensory input (Figure 3A), NA release is promoted by the sensory stimulus itself. It is also noteworthy that even low-frequency LC discharge (1–2 Hz) sharpened sensory responsiveness and receptive fields at the level of the thalamus and cortex [97,98,99]. Through depolarizing thalamic neurons, the LC also suppresses the appearance of sleep spindles that limit sensory throughput in thalamocortical areas [78]. The LC could hence promote sequential arousal-promoting actions that are graded with its activity levels as the transition from sleep to wakefulness takes place.

### 4.2. The LC as Part of the Regulatory Mechanisms of NREM Sleep

The real-time dynamics of NA for the first few hours of the light phase, the predominant resting phase of rodents, underscore the importance of the LC in the regulation of sleep architecture [72]. The elucidation of these dynamics across the light–dark cycle and across major sleep–wake control areas will reveal the full impact of NA on sleep’s brain states and associated sleep–wake behaviors. The LC is part of arousal circuits that are under circadian control [100] and it receives afferents from hypothalamic preoptic areas involved in NREM sleep homeostasis [31]. Therefore, beyond its regulation of sleep architecture and spectral composition, the LC could also contribute to circadian and homeostatic regulation of NREM sleep (Figure 3B).

### 4.3. The LC in REM Sleep Control

In spite of much pioneering work (see Section 2.1 and Section 2.2), how LC regulates REM sleep remains an open question (for review, see [101]). Recent research has focused on glutamatergic and GABA-ergic circuits involved in REM sleep regulation, whereas monoaminergic systems were attributed mostly a modulatory role (for review, see [102]). Measures of real-time NA dynamics, together with NREM sleep-specific optogenetic manipulation of the LC in rodents, instead indicate important changes in LC activity at moments of REM sleep onset. These recent data revive the questions about the LC’s role in REM sleep that we outline here in three aspects that could be important in future studies.

First, as described in Section 3, forebrain NA levels remained high during NREM sleep (Figure 2A) and LC neurons continue to be active [73]. This elevated activity in noradrenergic signaling suppresses REM sleep, as suggested by electrical or pharmacological LC stimulation in the rat [103,104] and by NREM sleep-specific optogenetic activation of noradrenergic LC neurons at low frequency [72]. A noradrenergic inhibition of REM sleep-promoting brain areas is a likely underlying mechanism for this suppression [105,106,107]. Importantly, the spontaneous activity of LC during natural undisturbed sleep seems even sufficient to antagonize NREM-to-REM sleep transitions. This was concluded from NREM sleep-specific optogenetic inhibition of the LC in freely sleeping mice, which increased the time spent in REM sleep [72]. This indicates that the LC is a powerful target to manipulate the balance between NREM and REM sleep in response to various regulatory and experience-dependent processes (see below and Section 5.2).

Second, NA levels declined in both the thalamus and cortex in REM sleep (Figure 2B). This decline took tens of seconds to complete once REM sleep began, raising the question of which are the determinants of this time course. The LC is inhibited by GABAergic mechanisms [108,109], of which several have been tested for their role in REM sleep control. Monosynaptic inhibitory afferents arise from the local and pericoerulear interneurons [110], ventrolateral periaqueductal gray [111], and from *nucleus prepositus hypoglossi* and dorsal paragigantocellular reticular nucleus [107,112,113]. Acetylcholine release from cholinergic REM sleep-promoting areas could also act through GABAergic mechanisms [109]. Cholinergic areas are likely initiating the LC inhibition as their discharge onset precedes REM sleep [84], but auto-inhibitory mechanisms within the LC could also play in at this moment [114]. At least one of the dorsal medullar inhibitory afferents increases activity exactly at REM sleep onset [113], suggesting that NA decline could become strengthened due to additional sources of inhibition. NA uptake mechanisms lagging behind synaptic inhibition of the LC could instead retard the decline of free NA levels. How the strength and the efficiency of synaptic inhibition regulate LC silencing and NA decline and/or interact with other excitatory and/or modulatory synaptic mechanisms of LC inhibition (see e.g., [110]) is currently unexplored. The determinants of NA dynamics at NREM-to-REM sleep transitions are critical to understanding how REM sleep evolves into an NA-free state because of its likely role in the regulation of emotional memory (see Section 5.2).

Third, the fluctuating levels of NA during NREM sleep indicate that chances for a NREM-to-REM sleep transition increase at moments of relatively low NA levels. Interestingly, the probability to enter REM sleep was indeed found to be phase-locked to the infraslow fluctuation of sigma power measured at the level of the EEG [113]. This lends support to the idea that fluctuating LC activity during NREM sleep generates brain states that are permissive for transitions, such as the ones to REM sleep (Figure 3A) [75,76]. The LC activity oscillating between high and low levels might suppress REM sleep on the one hand, but also open moments where transitions are favored. The LC is, therefore, positioned as a brain area capable of autonomously regulating the timing of REM sleep in a bidirectional manner during NREM sleep, yet how it is integrated into REM sleep regulatory mechanisms will require further research.

Alterations in REM sleep propensity, duration, and hippocampal-related theta activity are ubiquitous after stress- and fear-related experiences. These are part of the acute physiological responses to the hormonal and autonomic changes accompanying stress [115], but they also contribute to the consolidation of fear- [116] and extinction-related memories [115]. Increases in REM sleep are part of an adaptive process to mild stress exposure [117]. Given the LC’s high reciprocal connectivity with areas implied in fear, such as the amygdala (for review, see [31]), it is a strong candidate for linking stress-related experiences during the day to the timing of REM sleep. Indeed, acute decreases in REM sleep in response to inescapable footshock could be alleviated by optogenetic inhibition of excitatory neurons in the basolateral amygdala [118] or by dual hypocretin receptor antagonism in the LC or the dorsal raphe [119]. In case of such or even more traumatic experience, states of hyperarousal associated with elevated monoaminergic activity may arise, to which the LC contributes (for review, see [120]) (see Section 5.2).

### 4.4. The LC in Hippocampus-Dependent and Independent Memory Consolidation

LC activity, in part due to its implication in novelty detection, has been found to actively contribute to online memory consolidation. A series of studies found that activation of the LC favors different types of learning such as spatial learning [121,122,123,124], fear learning and reconsolidation [124,125,126], and perceptual learning [127,128]. Optogenetically activating LC tyrosine hydroxylase-positive neurons shortly after memory encoding of food rewards in a navigation task promoted memory retention in mice, which persisted until the next experimental assessment [121]. Such LC stimulation mimicked the effects of environmental novelty on memory encoding. Intriguingly, local pharmacological inhibition of dopaminergic but not noradrenergic receptors in the hippocampus implied a role of LC fiber-dependent dopamine release in novelty enhancement of hippocampus-dependent memory. Optogenetic stimulation of the LC during a spatial object recognition task lead to similar results [122]. Inhibition of the LC had, on the contrary, a detrimental effect on hippocampal place cell formation in goal-directed spatial learning [123]. The LC’s role in cued fear conditioning concerned both, memory acquisition of the pairing between the conditioned and the unconditioned stimulus, and later extinction [126]. Here, a dual role for LC afferent projections to the amygdala and to the medial prefrontal cortex could be identified, with the former implied in the acquisition, and the latter in the extinction of fear memory, demonstrating a modular functionality of LC subgroups depending on their projection targets. Pairing LC activation with stimulus presentation could also accelerate the learning of a new target sound in a perceptual learning paradigm [127] in rats and electrical/optogenetic stimulation of the LC during sound presentation promoted NA-dependent long-term plastic strengthening in auditory tuning curves of primary auditory cortex neurons [128].

The LC’s role as a regulator of memory acquisition likely relies on manifold actions of NA on neuronal excitability, in particular in hippocampal circuits, and on enduring changes in synaptic strength (for review, see [129]). One principal action of endogenously released NA, identified through optogenetic stimulation of LC fibers, appears to be a suppression of postsynaptic potassium currents, which enhanced the excitability of CA1 pyramidal neurons in response to Schaffer collateral stimulation [130]. This effect was blocked by β adrenoceptor antagonists, with no apparent implication of dopamine release. It is noteworthy that this action was already present when fibers were stimulated at low frequency (1 Hz), suggesting that such neuromodulation could be effective during NREM sleep, when the LC discharges at low frequencies (see Section 2.1).

In contrast to the strong evidence for the LC’s involvement in the memory acquisition phase, evidence that it plays a role during offline processing, including during sleep, is currently scarce. Pioneering pharmacological studies found that rats trained in an olfactory reward association task performed less well when they were injected with adrenergic antagonists intracerebroventricularly [131] or within prefrontal cortex [132] 2 h after training, but not at shorter or longer time intervals. These authors also provided evidence for a transient increase in NA levels during the time window in which these antagonists were effective. This pointed to a delayed re-activation of the LC that facilitated offline processing and memory consolidation. Follow-up studies suggest that such re-activation of the LC may indeed occur during post-learning sleep stages, as LC unit activity transiently doubled within the presumed re-activation window, without apparent alteration in sleep architecture [44]. The activity of LC units was further observed to be time-locked to slow waves in both rat [45] and human [64] and to hippocampal sleep spindles [46], suggesting that enhanced NA release is linked to the sleep rhythms that enable active systems consolidation. Finally, high-frequency stimulation of the LC disrupted the coupling of sleep spindles with hippocampal ripples that are high-frequency oscillatory patterns critical for memory consolidation [133]. This adds to evidence that the degree of LC activity might be critical in coordinating sleep rhythms relevant for offline processing (for review, see [134]).

### 4.5. The LC as Mediator of Vagal Afferent Information

Among the innervations that the LC receives, one is of particular interest as a gate for interoceptive signals, the Nucleus Tractus Solitarius [135,136] (for review, see [137]). This brainstem nucleus is part of the dorsal vagal complex (nucleus tractus solitarius, *area postrema* and dorsal motor nucleus of the vagus) which is the first recipient for vagal afferents (for review, see [138]). The vagus nerve is part of the parasympathetic system and it is a mixed nerve containing both motor and sensory fibers. Sensory information arising from the vagus nerve is important for autonomic feedback reflexes, such as the baroreceptor reflex and the Hering–Breuer reflex that serves to control breathing (for review, see [139]), and it reaches the LC via the dorsal vagal complex [140]. Vagus nerve stimulation is well-known for its beneficial role in clinical conditions, as evident from the highly diversified effects of vagus nerve stimulation (VNS). Indeed, this technique has been proposed to facilitate brain plasticity [141] (for review, see [142]) and memory formation (for review, see [143]). Some important domains of clinical application for VNS include drug-resistant epilepsy [144] (for review, see [145,146]), depression [147] (for review, see [148]), eating disorders [149], and neurodegenerative disorders [150].

Several animal studies support the LC as a major target of vagal afferent nerve stimulation. VNS caused an increase in the expression of the immediate-early gene c-*fos* in LC neurons in conscious unanesthetized rabbits [151] and in anesthetized rats [152]. Moreover, lesioning of the LC led to a suppression of the anticonvulsant effects of VNS in epileptic rats, supporting the idea that the LC is involved in this circuitry [153]. This implication of the LC was further supported by directly recording LC unit activity during VNS [154,155,156,157]. Using in vivo Ca${}^{2+}$ imaging in head-fixed awake mice, a recent study showed an increase in the noradrenergic neuromodulatory system in response to VNS [158]. Furthermore, in vivo microdialysis showed an increase in NA extracellular levels in the hippocampus and cortex during chronic VNS in anesthetized rats [159,160] and an increase in dopamine in extracellular levels in the prefrontal cortex and nucleus accumbens [161]. Additionally, vagal afferent electrical stimulation has been related to pupil dilation in animals and humans [162,163,164,165], consistent with the correlation between pupil diameter and firing of noradrenergic LC cells (for review, see [7]). Together, these results indicate that monoaminergic systems, including the LC, act as monitors of internal stimuli conveyed by vagal afferents (Figure 3A).

Given the role of the LC in the regulation of sleep, stimulation of vagal afferents may contribute to LC-dependent sleep regulatory effects. Animals studies suggest that VNS can promote REM sleep [166,167] and/or increase NREM sleep quantity as well as power in the delta and sigma bands [168] in freely moving cats. Several clinical studies also investigated the effects of VNS on sleep regulation. In epileptic and depressive patients, VNS treatment improved daytime alertness [169], increased the mean sleep latency [170], decreased awake time and stage 2 sleep and increased stage 1 sleep [171], increased delta power during NREM sleep and reduced REM sleep quantity [172,173], increased time spent in NREM sleep and decreased sleep latency and stage 1 sleep [174], and increased wakefulness and decreased light sleep and REM sleep [175]. These differences in the outcome could be related to the variability of the VNS parameters and/or the use of antiepileptic drugs which are known to affect sleep architecture (for review, see [176]).

So far, the contributions of sensory and motor components of VNS to sleep have not been determined. In a first step in this direction, a chemogenetic stimulation of the sensory afferents of the vagus nerve showed an alteration of sleep architecture and spectral composition, and a strong increase in the latency to REM sleep onset [177].

### 4.6. The Role of the LC in the Regulation of Brain Vascular Activity

DBH-positive LC terminals are tightly apposed on the fine arborizations of the neurovascular tree, notably the intraparenchymal capillaries. There is also evidence that released NA regulates cerebral blood flow, neurovascular coupling, and the maintenance of the blood–brain barrier (for review, see [178]). For example, the localized increase in blood supply to the somatosensory cortex, in response to paw stimulation depended on an intact LC [179,180].

As NA levels remain high in the forebrain during NREM sleep, it is likely that its actions on the microvasculature continue (Figure 3B). The LC innervates several components of the neurovascular unit, including astrocytic endfeet, as well as peri- and endocytes, which control different aspects of glial and capillary function (for review, see [181]) that are regulated differentially between sleep and wake [73]. One of the most important insights in this field was obtained for the brain’s glymphatic system that regulates the entry of cerebrospinal fluid along the perivascular space of small capillaries (for review, see [182]). Fluid exchange via the glymphatic system is enhanced during NREM sleep and cleanses the brain from toxic products such as amyloid-β-protein [183]. The fluctuating NA levels during NREM sleep could hence contribute to the pulsatile nature of this exchange process, perhaps through acting on vasomotor activity that is thought to be critical for the paravascular clearance of solutes, in particular when occurring at infraslow frequencies [184]. Interestingly, a recent study indicated a temporal correlation between cerebrospinal fluid exchange and the occurrence of slow and infraslow electrical activity in the EEG [185]. In view of these most exciting developments, we speculate that the LC’s dual capability of modulating neural oscillation control and arteriolar vasoconstriction makes it a master regulator of the sleeping brain’s functions because it could potentially play a role in coordinating the timing of sleep architecture, sleep electrical rhythms, and brain waste clearance.

An implication of the LC in gross cerebral blood flow arises from functional MRI studies. These have repeatedly reported the presence of spontaneous slow signal fluctuations during rest and sleep, including during N2 sleep in humans. Frequencies involved are in the infraslow range, close to values found for infraslow activity fluctuations of the LC during NREM sleep in rodent [186,187,188]. Furthermore, chemogenetic activation of the LC in lightly anesthetized mice generates a functional activation pattern [33] that overlaps with some of the areas found in early sleep stages [188]. The infraslow activity of the LC during NREM sleep could conceivably impose a time frame for resting-state network activity, which remains a question for future work.

## 5. The LC and Sleep Function in Pathology

As the LC has been known primarily as a wake- and attention-promoting brain area, the idea that LC dysfunctions could play a role in sleep (rather than wake) problems has been less considered. Moreover, the idea that a dysfunctional LC could be involved in a decrement of some major neuroprotective roles of sleep is so far underexplored. As the LC’s profound implication in sleep architecture and sleep function is increasingly recognized, these possibilities come to center stage and open novel inroads for preventive strategies (Figure 3B).

### 5.1. Aging and Neurodegenerative Disorders

Many aspects of sleep, from its timing and initiation to its maintenance and depth deteriorate with aging (for review, see [189]), and this process is aggravated in the case of neurodegenerative dementias, of which Alzheimer’s disease (AD) is the most common form (for review, see [190]). In healthy aging mice, hypothalamic orexin neurons undergo increases in intrinsic excitability that cause sleep fragmentation [191]. In aging accompanied by neurodegeneration, much interest has recently focused on the LC that appears to be afflicted at early stages of AD [192]. Ample evidence further indicates that disturbed sleep adversely affects the progression of AD pathology (for review, see [193]). Therefore, addressing whether early LC pathology links to sleep disruptions bears potential to identify early stages of disease. This potential is strengthened by newest evidence that structural measures of LC integrity in vivo can be related to the initial stages of AD-related neurodegeneration and cognitive decline [194].

It is currently open how exactly LC neuronal activity and NA signaling are altered with aging and pathologically aggravated with the progression of AD. Chemogenetically stimulating LC in a rat model of AD recovered spatial learning capacities, but how much and in which brain areas NA signaling was restored remained an open question [195]. As free NA dynamics have become accessible through biosensors (see Ch. 3), it is now possible to determine when and how these are affected by the neurodegenerative processes and to which types of sleep disruptions they might be linked. Amongst the diverse alterations in sleep in patients with neurodegenerative disorders (for review, see [190]), recent focus has been on alterations in sleep’s microarchitecture [66,196] and possible links to LC dysfunction, which make altered NA signaling during NREM sleep as a reasonable path to be pursued. On top of this, evidence for the LC’s implication in the vascular pathology and decline of glymphatic activity in AD pathogenesis has attracted enormous interest (for review, see [178]). At this stage, deepening the causal links between LC dysfunction and altered NA signaling is a very promising path to the LC’s broad implication in sleep disorders linked to neurodegenerative diseases (for review, see [36,182].

### 5.2. Stress-Related Disorders

Increased noradrenergic LC activity is a common observation after stressful or traumatic life experiences (for review, see [25]). This increase persists beyond the momentary insult and may continue during sleep. Even comparatively mild stress in rats, such as a simple cage exchange, activates major wake-promoting areas, including the LC, and leads to sleep fragmentation [197]. Both mild and excessive stress, such as the one inflicted by traumatic events, have been related to a maintained hyperactivity of the LC noradrenergic system (for review, see [25]). As stress and various sleep disruptions are tightly linked, it is likely that the NA signaling profile during NREM and REM sleep becomes altered at various levels and adversely affects sleep physiology.

First, elevated LC activity and NA signaling is arousal-promoting through its desynchronizing effect on EEG that favors high- over low-frequency oscillatory activity, as demonstrated by pharmacologic [198], electrical [199], chemogenetic [200], or optogenetic [63] activation of LC neurons. Alteration in the LC noradrenergic system may thus contribute to cortical hyperarousal states during sleep. Interestingly, cortical hyperarousal states are a common trait of sleep disruptions arising from neuropsychiatric conditions, but also from pain (for review, see [201]) and primary insomnia (for review, see [202]).

Second, elevated LC activity promotes arousability to external stimuli (see Section 4.1), facilitating sleep disruptions. It is well accepted that lightened NREM sleep and more frequent awakenings are part of the disease profile in post-traumatic stress disorder (for review, see [25,203]).

Third, elevated LC activity may compromise the decline of NA levels during REM sleep. While this possibility awaits a direct demonstration, the idea that insufficient decline of NA levels during REM sleep has been put forward as a mechanism inhibiting extinction of emotional memory (for review, see [25,202]). Mechanistically, it is thought that the quiescence of LC neurons during REM sleep allows a depotentiation of synaptic strength in anxiety-related networks, including the amygdala. Therefore, during NA-enriched REM sleep, also referred to as “restless REM sleep”, behavioral reactions to emotional stress do not decline overnight [204].

More generally, high and fluctuating levels of NA in NREM sleep may support synaptic plasticity while the low levels during REM sleep could promote synaptic depotentiation and downscaling. As a consequence, aberrant noradrenergic activity during REM sleep may contribute to the maladaptive recall of complex experiences in which emotional aspects remain highly salient. The real-time dynamics of NA during NREM and REM sleep will be essential in refining the proposed picture of the LC as an important coordinator of memory consolidation processes during sleep.

### 5.3. Sleep and Cardiovascular Regulation

The cardiovascular correlates of NREM and REM sleep arise from the interplay of autonomic reflex arcs and central commands that regulate the balance between sympathetic and parasympathetic activity (for review, see [205]). Both circadian and sleep-driven mechanisms contribute to the central control of the cardiovascular system (for review, see [206]). NREM sleep is dominated by parasympathetic influences, whereas sympathetic ones prevail in REM sleep (for review, see [206,207]). LC efferents target both preganglionic sympathetic and parasympathetic output areas, activating the former while inhibiting the latter. Further cardiovascular impact may arise through the LC’s connections with stress- and attention-responsive brain areas (for review, see [29]). However, the LC’s role in the central autonomic commands for cardiovascular control in sleep is not clarified, although brainstem mechanisms are particularly prevalent in cardiovascular control during NREM sleep (for review, see [207]). In mice, infraslow variations in heart rate during NREM sleep were mediated by the parasympathetic system [72]. Furthermore, continuous and global optogenetic stimulation of LC noradrenergic neurons during NREM sleep disrupted previously observed anticorrelations between spindle clustering and heart rate, whereas LC stimulation at infraslow frequencies strengthened this anticorrelation [72]. The LC is thus positioned to regulate central and autonomic activity during NREM sleep. Given the numerous sleep-related cardiovascular alterations in neuropsychiatric and neurodegenerative diseases, it will be of great interest to examine the LC’s and other monoaminergic’s contributions to the pathophysiological manifestations of these conditions [207].

## 6. Closing Remarks and Future Directions

We outlined novel evidence showing that the noradrenergic LC plays important and previously underestimated roles in sleep. We reviewed and contrasted existing literature with recent findings that unraveled the real-time dynamics of the LC and its NA output during sleep.

A central step forward is the recognition that NA signals span an unexpectedly high dynamic range, from high and comparable levels between wakefulness and NREM sleep to low levels in REM sleep, at least in the two forebrain areas measured so far. This dynamic currently is not congruent with what we know about variations in LC unit activity across sleep and wakefulness. Clearly, much is still unknown about how LC neuronal activity determines NA release, possible target-specific presynaptic release properties, and variations in local uptake mechanisms, all of which shape NA dynamics. It is furthermore going to be important to determine whether these fluctuations arise as part of the LC’s spontaneous activity and/or secondarily from its integration into large-scale sleep-regulatory networks within the central and autonomic nervous systems. In this review, we outlined that recognizing NA as a neuromodulator during sleep opens novel mechanistic ideas on how sleep architecture and spectral dynamics are organized to the benefit of sleep functions. Future studies will undoubtedly reveal that fluctuations in other neuromodulators, such as the ones already reported for serotonine [86] and dopamine [92], work conjointly with NA in these processes.

An additional unique observation is the infraslow fluctuations in NA levels that characterize NREM sleep. These dynamics bring, for the first time, a neural in vivo foundation to a time scale of brain oscillatory activity that has long revolved in whole-brain measures and behavioral output, but that has not been a systematic part in the check-box list of sleep rhythms that are important for sleep functions [77]. Now, times become ready for speculations about its origins in the coordination of sleep and offline brain functions that are central to brain and bodily health.

As they currently stand, these new observations will have manifold implications for the LC’s role in healthy and disordered sleep. Some of these implications have been proposed but not pursued for years, yet they are now accessible with unprecedented spatiotemporal control. Most intriguingly, we may soon come to realize that the high NA levels are integral to enabling restorative NREM sleep and generating its unique benefits for health. Some other implications, however, are newly emerging. The NA, and perhaps other monoamines, present a profile of sleep as a behavioral state that integrates neuromodulation to monitor environmental, bodily, and brain states to enable adaptive behaviors. We propose that NA could show us the way to the neural foundation of a vigilance system for sleep, based on which novel insights into sleep’s benefits and in-roads for therapeutic treatments of sleep disorders arise.

## Figures and Tables

**Figure 1 ijms-23-05028-f001:**
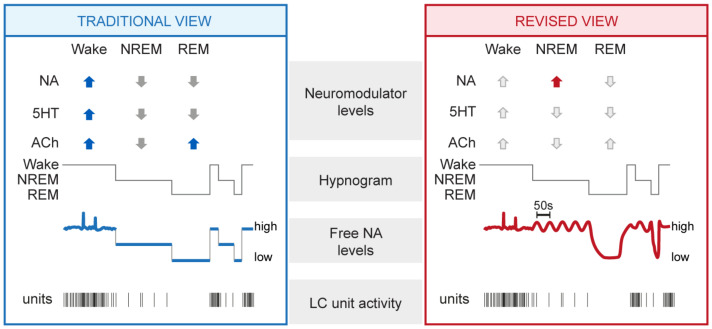
Summary of traditional and revised views on the neuromodulatory profiles of wakefulness and sleep, with a focus on noradrenergic signaling. Traditional (**left**) and revised (**right**) views derived from animal experimentation are summarized and complemented with data-derived schematic representations of NA dynamics and LC unit activity. From top to bottom: mean levels of major neuromodulators (blue up and gray down arrows symbolize high and low levels in the traditional view), a representative hypnogram of mouse sleep–wake behavior, free NA levels, and representative discharges of a LC unit. Novel insights central to the revised view are highlighted with the red arrow, whereas unaltered neuromodulatory levels are shown with light grey arrows. NA, noradrenaline; 5HT, serotonin; ACh, acetylcholine; NREM, NREM sleep; REM, REM sleep.

**Figure 2 ijms-23-05028-f002:**
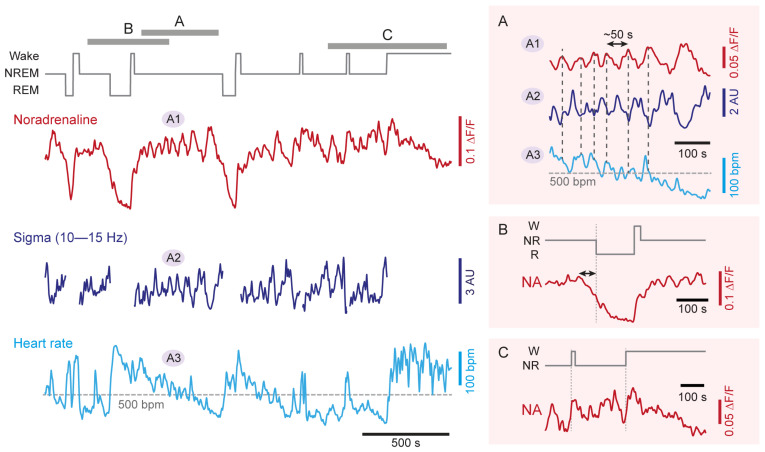
Real-time dynamics of NA levels in somatosensory thalamus, forebrain sleep spindle power, and heart rate during NREM sleep. Representative simultaneous recordings in a freely behaving mouse combining (from top to bottom): hypnogram (*gray*), free NA levels in somatosensory thalamus obtained through fiberphotometry imaging (*red*, A1), local field potential sigma power (10–15 Hz) in somatosensory cortex (*dark blue*, A2) and heart rate (*light blue*, A3), with labeled portions (A, B, C) shown expanded on the right. The variations in sigma power reflect the clustering of sleep spindle density ([72]). Insets on the right expand portions of the traces highlighted with letters in the hypnogram to show (**A**) NREM sleep (double-headed arrow marks the 50 s periodicity); (**B**) NREM-to-REM sleep transitions (double-headed arrow marks the decay time of NA levels prior to REM sleep onset); (**C**) NREM-to-wake transitions. Portions of two of these traces have been published previously [72]. NA, noradrenaline; W, wakefulness; NR, NREM sleep; R, REM sleep; ΔF/F, relative fluorescence changes; AU, arbitrary unit; bpm, beats per minute.

**Figure 3 ijms-23-05028-f003:**
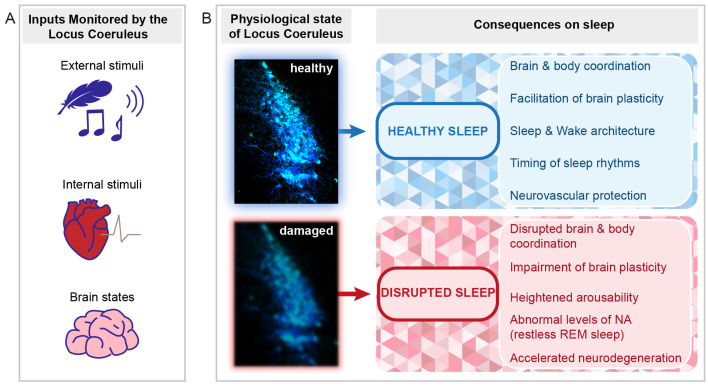
Perspectives for the implication of the LC in healthy and disrupted sleep. Schematic indicating the types of signals monitored by the LC and the implications of LC function and dysfunction for sleep. (**A**) The LC monitors external stimuli (e.g., sensory stimuli such as touch or sound, symbolized by a feather and a musical note, see Section 4.1), internal stimuli (symbolized by the heart, see Section 4.5), and internal brain states important for the regulation of NREM-to-REM sleep transitions (symbolized by the brain, see Section 3.4 and Section 4.3). (**B**) Depending on the LC status (healthy or damaged), beneficial or adverse consequences on sleep can arise. Several outcomes are listed on the right. The LC micrograph was obtained from an immunohistochemically stained brain section of one of the mice used for the data published in [72]. The color choice of cell labeling was made deliberately to mark it as the sky-blue spot. The blurring of the blue color in the bottom micrograph symbolizes both structural and functional alterations that can lead to LC dysfunction.

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
