# Peer review of "When the Locus Coeruleus Speaks Up in Sleep: Recent Insights, Emerging Perspectives"

_ijms, 2022, doi:10.3390/ijms23095028_

Round 1
Reviewer 1 Report
I have the honor to review the paper by Osorio-Forero et al. Authors take an interesting topic of recent insights and emerging perspectives in the context of dependencies between locus coeruleus and sleep. This is a comprehensive review of 190 papers. The authors provided high-quality Figures, which make the text easier to read. Very good aspects of the proposed manuscript are references to reviews devoted to more detailed parts of this topic. Nevertheless, I have a few comments which should be addressed before publication:
Major
- chapter 1 title should be changed
- references should be stated at the end of the sentence if it is possible (e.g. line 88)
Minor
- the affiliation should be stated in a detailed manner
- In line 57 please add "Van Egroo et al." before [36]
Author Response
NOTE: THIS REVIEWER WAS THE ORIGINAL REVIEWER 3
I have the honor to review the paper by Osorio-Forero et al. Authors take an interesting topic of recent insights and emerging perspectives in the context of dependencies between locus coeruleus and sleep. This is a comprehensive review of 190 papers. The authors provided high-quality Figures, which make the text easier to read. Very good aspects of the proposed manuscript are references to reviews devoted to more detailed parts of this topic.
Our response: We are grateful for this reviewer’s positive assessment of our review.
Nevertheless, I have a few comments which should be addressed before publication:
Major
- chapter 1 title should be changed
Our response: As suggested, we have now changed this title and aligned it with the one of Chapter 2.
- references should be stated at the end of the sentence if it is possible (e.g. line 88)
Our response: We do cite the references at the end of the sentences whenever possible. We think that on line 88 (now line 101) it is important to cite ref 51 (now Ref 53) in the middle sentence to ensure that it relates to Jouvet’s work and not to the examinations that were prompted by his theory.
Minor
- the affiliation should be stated in a detailed manner
Our response: We have now completed the affiliation of all authors.
- In line 57 please add "Van Egroo et al." before [36]
Our response: This has been done on line 61.
Reviewer 2 Report
The Locus Coeruleus (LC) and noradrenergic system have always a great interest given their central role on brain states and cognition. here Osorio-Forero et al. performed and extensive review on any evidence relating the LC to sleep.
Overall, this is a thoroughly researched review which covers all essentials aspects ranging from gerneral interactions/dynamics to the implications for clinical conditions. It also refers to other reviews which I highly appreciate.
As a sleep researcher myself I find this work an great collection of relevant literature. However, this brings me to my only concern: given that I do not have a strong molecular background, a few sections are highly technical and thus difficult to read (e.g. "Is the LC active in sleep? Novel insights, page 4, lines 141-162). Furthermore, often "technical terms" were dropped without much context (e.g. on page 6 suddenly hippocampal theta and ripples are named). Depending on the targeted readership, it might be helpfuly to ease into a few aspects and reduce the technicality.
Finally, I would have personally liked a central figure illustrating the location of the LC within the brain and perhaps also the structures/organs it connects to in order to help understand all the descriptions in the main text.
Author Response
NOTE: THIS REVIEWER WAS THE ORIGINAL REVIEWER 1
The Locus Coeruleus (LC) and noradrenergic system have always a great interest given their central role on brain states and cognition. here Osorio-Forero et al. performed and extensive review on any evidence relating the LC to sleep.
Overall, this is a thoroughly researched review which covers all essentials aspects ranging from gerneral interactions/dynamics to the implications for clinical conditions. It also refers to other reviews which I highly appreciate.
Our response: We appreciate this positive assessment of the reviewer. In the revised version, we explicitly cite many outstanding reviews on LC at key sites, and we make sure that these are clearly highlighted by adding ‘for review, see..’ systematically throughout the text.
As a sleep researcher myself I find this work an great collection of relevant literature. However, this brings me to my only concern: given that I do not have a strong molecular background, a few sections are highly technical and thus difficult to read (e.g. "Is the LC active in sleep? Novel insights, page 4, lines 141-162). Furthermore, often "technical terms" were dropped without much context (e.g. on page 6 suddenly hippocampal theta and ripples are named). Depending on the targeted readership, it might be helpfuly to ease into a few aspects and reduce the technicality.
Our response: It is our goal to target a broad readership with this review that includes both sleep researchers and neuroscientists interested in the functions of the LC. Therefore, in the revised version, we make ensure that technical aspects from both the sleep but also the neurobiology of the LC are introduced at appropriate sites. Specifically:
Beginning of Chapter 1 (lines 66-69):
We briefly introduce the basic concepts of the two major sleep states, NREM and REM sleep, and add a review article (Ref. 37).
In section 1.1 (line 83-88):
We introduce two major electroencephalographic rhythms of NREM sleep, the sleep spindles and the slow waves.
In section 1.1 (line 89-90 and 96-97, respectively):
We introduce in one sentence each the basic idea of the microdialysis and the voltammetry techniques.
In section 1.2 (lines 132-139):
We introduce in simple words the basic ideas of neuroimaging techniques in human and provide appropriate references (Ref. 36, 65) regarding their use for LC studies in human sleep.
Beginning of Chapter 2 (lines 158-172):
We introduce the technique of noradrenaline biosensors using more general wording and leaving out technical details indicated in the previous version.
In section 2.1 and 2.2:
We avoid technical jargon from the sleep field and provide explicit explanations to the traces used in Figure 2. We also make sure that the description of the original traces in Figure 2 is done appropriately in both the legend and the text (lines 205-221).
In section 3.4 (line 418-419):
We introduce the hippocampal ripples as an oscillatory pattern that is critical for memory consolidation.
With these revisions, we hope to make the review accessible for the broad readership mentioned.
Finally, I would have personally liked a central figure illustrating the location of the LC within the brain and perhaps also the structures/organs it connects to in order to help understand all the descriptions in the main text.
Our response: Not being experts ourselves in the functional anatomy of the LC and in view of the numerous excellent reviews written by experts, we propose to refer to these to help understand the anatomical and connectivity parts in the text. Accordingly, we provide several reviews on particular aspects of the functional anatomy of the LC in the introduction to the review (lines 42-50). We also provide additional reviews that include anatomical information in particular chapters, such as in the Section 3.3. (‘The LC in REM sleep control’) and in Section 3.5. (‘The LC as mediator of vagal afferent information’).
Reviewer 3 Report
There has been a burgeoning of studies of the locus coeruleus noradrenergic system in the past few years due to spectacular developments in biotechnology permitting access to this tiny brain stem nucleus. This has resulted in several recent publications of excellent, scholarly reviews of the LC literature providing new insights into its role in cognition and behavior. The present review article has the potential of making a significant, novel contribution, with its focus on sleep. However, careful scrutiny of the manuscript reveals many errors, especially in the literature citations.
There are many examples throughout the manuscript of incorrect citations. Just to focus on one section alone, 3.4:
-The opening sentence states that ‘interplay between LC and hippocampus….’. Neither of the two references cited support this statement. Ref 34 is a very general review paper on LC and Ref 92 addresses the role of LC in arousal from sleep, with no mention of hippocampus. In fact, since the hippocampus does not directly project to LC, any ‘interplay’ would have to be through intermediary structures.
-In the next sentence (line 364) ‘the role of LC in the encoding process has been demonstrated…’, none of the three papers cited addresses the role of LC in encoding. Ref 45 looks at off-line memory consolidation, not encoding.
-Line 372: ref 122 does, indeed, show the importance of noradrenergic input to the lateral amygdala for fear memories but does not show that this depends on interaction with the hippocampus.
-Line 374 ‘…LC stimulation can suppress hippocampal sharp wave ripples…’ ; the reference cited 118, actually showed that LC high frequency stimulation suppressed ripple-spindle coupling and it was this interference with hippocampal cortical dialogue, according to the authors, that accounted for the memory deficit, not suppression of hippocampal ripples.
-Line386 The experiments described in ref 43 used rats, not mice.
-Line 398 ref 125 ’…antidromic stimulation…’ that study actually showed that a small population of PFC neurons were antidromically activated by LC stimulation.
--Line 400 The importance of interaction between LC and cortex during sleep is suggested by ref 118.
-Line 393 It has been known for many years that NE action mediated through beta receptors was essential for hippocampal LTP,in vitro (e.g. Stanton & Sarvey, 1985) and in vivo.(e.g. Babstock & Harley, 1992; Klukowski & Harley, 1994). See also many review papers.
Thus, one short section of the proposed review, when carefully scrutinized, shows many errors in citation and interpretation of the literature. This scrupulous control has not been applied to the rest of the manuscript; similar lapses of scholarship may appear throughout. As a general principle, good scholarship demands that cited papers should be read by the authors to assure the relevancy of the reference and the accuracy of the presentation.
Minor points:
Line 284 word missing?
Line 287 rodents
Line 313 what do you mean by Consolidates NREM sleep? Also line 244 ‘when NREM sleep appeared consolidated’.
Comments on figures:
Is fig 1 showing real data (in the ‘free NE levels’ row of the figure) or is it a cartoon? Need more information on this figure.
Fig3 This figure is very confusing. It looks like the external stimuli are inputting to ‘healthy LC’, internal stimuli nowhere and brain states to ‘damaged LC’. The ‘healthy ‘LC image is a published immuno section, but what is the lower ‘damaged’ image? How was that produced?
This cartoon figure is confusing and does not add anything to understanding the point of the review.
To sum up --this massive review has the potential of being a significant contribution to the literature, but it requires careful editing, pruning and attention to detail,especially in the accuracy of cigtations.
Author Response
NOTE: THIS REVIEWER WAS THE ORIGINAL REVIEWER 2
There has been a burgeoning of studies of the locus coeruleus noradrenergic system in the past few years due to spectacular developments in biotechnology permitting access to this tiny brain stem nucleus. This has resulted in several recent publications of excellent, scholarly reviews of the LC literature providing new insights into its role in cognition and behavior. The present review article has the potential of making a significant, novel contribution, with its focus on sleep. However, careful scrutiny of the manuscript reveals many errors, especially in the literature citations.
Our response: We acknowledge the reviewer’s recognition of the potential of our review. We also recognized the imprecisions in the citation of literature. We have now scrutinized the entire review for appropriate citations that notably includes an appropriate mentioning of the leading reviews on the different aspects of LC function in cognition and behavior. Moreover, we have fully rewritten the Section 3.4. For this section, the reviewer found a high number of incorrect citations. The first paragraph of this rewritten Section treats the role of LC in online memory consolidation. The second paragraph briefly reviews the role of LC in cellular mechanisms that support plasticity. The third paragraph treats the literature on the LC in offline consolidation.
There are many examples throughout the manuscript of incorrect citations. Just to focus on one section alone, 3.4:
-The opening sentence states that ‘interplay between LC and hippocampus….’. Neither of the two references cited support this statement. Ref 34 is a very general review paper on LC and Ref 92 addresses the role of LC in arousal from sleep, with no mention of hippocampus. In fact, since the hippocampus does not directly project to LC, any ‘interplay’ would have to be through intermediary structures.
Our response: The cited part of the text has been removed. Ref 34 (now Ref 35), which the reviewer correctly describes as ‘a very general review paper’ is no longer cited in this section. Ref 92 (now Ref 95), which the reviewer correctly describes as addressing ‘the role of LC in arousal from sleep’, is now cited in the Section 3.1. (line 274) that treats the role of the LC in sensory arousal.
-In the next sentence (line 364) ‘the role of LC in the encoding process has been demonstrated…’, none of the three papers cited addresses the role of LC in encoding. Ref 45 looks at off-line memory consolidation, not encoding.
Our response: This part of the text has been removed. The three papers that the reviewer mentions (Ref 117-119) have been re-cited in the following contexts. Ref 117 has been removed as it does not treat a topic relevant to the rewritten section. Ref 118 (now Ref 133) is now discussed in the part on off-line memory consolidation on line 420. Ref 119 (now Ref 134) is cited as the major review article of this section on line 421. Ref 45 (now Ref 46) is now cited in the paragraph that discusses the literature on off-line memory consolidation on line 416.
-Line 372: ref 122 does, indeed, show the importance of noradrenergic input to the lateral amygdala for fear memories but does not show that this depends on interaction with the hippocampus.
Our response: Ref 122 (now Ref 125) is now cited in the context of fear learning and consolidation without reference to the hippocampus on line 375.
-Line 374 ‘…LC stimulation can suppress hippocampal sharp wave ripples…’ ; the reference cited 118, actually showed that LC high frequency stimulation suppressed ripple-spindle coupling and it was this interference with hippocampal cortical dialogue, according to the authors, that accounted for the memory deficit, not suppression of hippocampal ripples.
Our response: Ref 118 (now Ref 133) is now appropriately described in the third paragraph of this Section (line 420) that treats the literature on offline consolidation.
-Line386 The experiments described in ref 43 used rats, not mice.
Our response: Ref 43 (now Ref 44) is now appropriately described in the third paragraph of this Section (line 414) that treats the literature on offline consolidation.
-Line 398 ref 125 ’…antidromic stimulation…’ that study actually showed that a small population of PFC neurons were antidromically activated by LC stimulation.
Our response: This reference has been removed from this review.
--Line 400 The importance of interaction between LC and cortex during sleep is suggested by ref 118.
Our response: In the revised version of this Section, we focus on evidence that LC is implied in online memory acquisition and in offline processing, but we no longer discuss the exact circuits involved. Ref 118 (now Ref 133) is now cited in the third paragraph of this Section (line 420) that treats the literature on offline consolidation.
-Line 393 It has been known for many years that NE action mediated through beta receptors was essential for hippocampal LTP,in vitro (e.g. Stanton & Sarvey, 1985) and in vivo.(e.g. Babstock & Harley, 1992; Klukowski & Harley, 1994). See also many review papers.
Our response: We thank the reviewer for the helpful indication of these important studies. In the revised version of this review that treats the role of the LC in sleep, we have decided to focus on LC’s role in memory-related functions. For cellular mechanisms, we instead rely on a recent excellent review by Palacios-Filardo et al. (Ref. 129) that we cite on line 396. We additionally cite a recent paper that looks at the effects of endogenous release of NA on hippocampal plasticity (Ref. 130) that we cite on line 399. We chose to cite this in vitro study because it includes results on the effects of 1 Hz-LC stimulation, which could be relevant for NREM sleep.
Thus, one short section of the proposed review, when carefully scrutinized, shows many errors in citation and interpretation of the literature. This scrupulous control has not been applied to the rest of the manuscript; similar lapses of scholarship may appear throughout. As a general principle, good scholarship demands that cited papers should be read by the authors to assure the relevancy of the reference and the accuracy of the presentation.
Our response: We are thankful to the reviewer for his/her scrutiny and for the critical but nevertheless very constructive input. We think that the full rewriting of Section 3.4 has gained much from this reviewer’s comments and now lives up to the standards of good scholarship. We also have scrutinized the rest of the manuscript and we are confident that we now accurately cite the relevant literature.
Minor points:
Line 284 word missing?
Our response: We found the corresponding sentence (now line 291) to be complete.
Line 287 rodents
Our response: This has been corrected (now line 298).
Line 313 what do you mean by Consolidates NREM sleep? Also line 244 ‘when NREM sleep appeared consolidated’.
Our response: We apologize for the technical jargon that was not made clear. We have removed the corresponding text portions to avoid such lack of clarity for the non-expert reader.
Comments on figures:
Is fig 1 showing real data (in the ‘free NE levels’ row of the figure) or is it a cartoon? Need more information on this figure.
Our response: This missing information is now given in the legend to this figure.
Fig3 This figure is very confusing. It looks like the external stimuli are inputting to ‘healthy LC’, internal stimuli nowhere and brain states to ‘damaged LC’. The ‘healthy ‘LC image is a published immuno section, but what is the lower ‘damaged’ image? How was that produced?
This cartoon figure is confusing and does not add anything to understanding the point of the review.
Our response: This figure is now subdivided into two parts, A and B. In A, we refer to the signals monitored by the LC. In B, we present possible outcomes for sleep depending on the state of the LC. With these changes, we think that this Figure helps the reader to obtain an overview over the roles of LC for sleep. The “damaged” LC represents pathological conditions that can damage the LC in functional or structural aspects. This is clearly said in the last sentence of the legend to this figure.
To sum up --this massive review has the potential of being a significant contribution to the literature, but it requires careful editing, pruning and attention to detail,especially in the accuracy of cigtations.
Our response: Again, we are grateful for this reviewer’s time and expertise that helped us substantially to become aware of the weaknesses of the previous version. We are apologetic for the lack of scholarship that we have now much improved. We hope to now convince the reviewer that we provide a review that has the potential that the reviewer has generously mentioned.